# Presence and Multi-Species Spatial Distribution of Oropouche Virus in Brazil within the One Health Framework

**DOI:** 10.3390/tropicalmed7060111

**Published:** 2022-06-20

**Authors:** Sofia Sciancalepore, Maria Cristina Schneider, Jisoo Kim, Deise I. Galan, Ana Riviere-Cinnamond

**Affiliations:** 1Department of International Health, Georgetown University, Washington, DC 20057, USA; mcs368@georgetown.edu (M.C.S.); dgl32@georgetown.edu (D.I.G.); 2Health Emergency Department, Pan American Health Organization (PAHO/WHO), Washington, DC 20037, USA; kimjis@paho.org (J.K.); rivierea@paho.org (A.R.-C.); 3Institute of Collective Health Studies, Federal University of Rio de Janeiro, Rio de Janeiro 21941-901, Brazil

**Keywords:** Brazil, disease mapping, One Health, Oropouche virus, risk

## Abstract

Oropouche virus (OROV) is an emerging vector-borne arbovirus with high epidemic potential, causing illness in more than 500,000 people. Primarily contracted through its midge and mosquito vectors, OROV remains prevalent in its wild, non-human primate and sloth reservoir hosts as well. This virus is spreading across Latin America; however, the majority of cases occur in Brazil. The aim of this research is to document OROV’s presence in Brazil using the One Health approach and geospatial techniques. A scoping review of the literature (2000 to 2021) was conducted to collect reports of this disease in humans and animal species. Data were then geocoded by first and second subnational levels and species to map OROV’s spread. In total, 14 of 27 states reported OROV presence across 67 municipalities (second subnational level). However, most of the cases were in the northern region, within the tropical and subtropical moist broadleaf forests biome. OROV was identified in humans, four vector species, four genera of non-human primates, one sloth species, and others. Utilizing One Health was important to understand the distribution of OROV across several species and to suggest possible environmental, socioeconomic, and demographic drivers of the virus’s presence. As deforestation, climate change, and migration rates increase, further study into the spillover potential of this disease is needed.

## 1. Introduction

Oropouche virus (OROV) attained its name by first being identified in a patient living in the city of Vega de Oropouche in Trinidad. Since this initial case in 1955, Argentina, Bolivia, Brazil, Columbia, Ecuador, Panama, Peru, and Venezuela have all reported the presence of OROV [1]. This virus was detected in Brazil for the first time in 1960. Among the wild area bordering the construction site of the Belem–Brasilia highway, blood samples taken from a sloth (*Bradypus tridactylus*) and multiple mosquitoes (*Ochlerotatus serratus*) tested positive for OROV [2]. The following year, this virus demonstrated its high epidemic potential when it was detected in the capital of Pará state, where a major epidemic arose, causing illness in approximately 11,000 people [2]. OROV continues to cause further epidemics in different urban centers among the north and northeast regions of Brazil, specifically in the states of Pará, Amapa, Amazonas, Acre, Maranhão, Tocantins, and Rondonia [3].

OROV is the etiological agent for Oropouche fever, responsible for over half a million infections across Latin America and the Caribbean since the 1950s [4]. However, the majority of the cases reported have occurred in Brazil, where several serious outbreaks of Oropouche virus have taken place since its identification. Not only is OROV responsible for over half a million human cases in the country so far, but it is also steadily encroaching on new geographical boundaries [5]. 

Outbreaks of OROV with large case counts have been reported in Brazil since 1961 (Table 1). Between the years of 1961 and 1975, the state of Pará experienced four major outbreaks, resulting in over 131,000 cases in only 14 years. OROV then rapidly spread to the neighboring states of Amazonas and Rondonia, before reaching Mato Grosso state in the 2010s. These outbreaks clearly demonstrate the continued circulation of the virus within the country. Today, following dengue virus, OROV is considered to be the second most common arbovirus in the Brazilian legal Amazon region [4]. The results of this study reflect the historical distribution and magnitude of OROV and will further highlight key cases that mark the spread of OROV to new regions in Brazil.

Through spillover events facilitated by the bite of an infected midge or mosquito, OROV has become an increasing concern in human health. Spillover is defined as an event in which the presence of a disease moves from its animal host into a human case [6]. Patients typically present with febrile symptoms of fever, chills, photophobia, skin rashes, and dizziness [1]. 

If given the opportunity to spread, Oropouche virus can have staggering effects. While no fatalities associated with OROV have been reported to date, some patients have been known to experience serious health consequences as a result of this infection [7]. For instance, encephalitis and meningitis (brain swelling) have been reported, as well as instances of spontaneous bleeding [8]. This was seen during the outbreak in Pará state from 1979 addressed above, when 4.1% of these patients developed meningitis [4].

OROV is maintained in nature through two distinct cycles: an urban and sylvatic cycle. As an arbovirus, OROV’s urban cycle is primarily upheld by its midge vector (*Culicoides paraensis*), as well as some mosquito vectors (*Culex quinquefasciatus*, *Aedes aegypti, Ochlerotatus serratus)*. A midge is a small fly that has an average lifespan of 20 to 30 days. Notably, only adult, female midges are responsible for pathogen transmission. This is because only female vectors take blood meals to support egg production and maturation [4]. As it has historically been linked to several large-scale epidemics of the disease, this midge species is commonly considered to be the most important vector to spread OROV [4]. 

Additionally, *Culicoides* midges are considered to be of public health concern globally as they are a known vector species of several other arboviruses (such as Equine encephalitis and Schmallenberg virus), and approximately 96% of these midge species take blood meals from humans and wild mammals [4]. This is of importance as, in the urban cycle, OROV can be transmitted to individuals who are susceptible to the bite of an infected vector during the blood meal [3]. The *Culex quinquefasciatus* mosquito is also a considerable urban vector for OROV as it is widely found in tropical regions, where it takes blood meals from humans and animals [4]. To date, zero cases of human-to-human transmission of Oropouche fever have been reported.

In its sylvatic cycle, mammals such as non-human primates (NHPs) and sloths (*Bradypus tridactylus*) act as potential reservoir hosts for OROV. It has been found that the howler monkey (*Alouatta caraya* and *Alouatta guariba clamitans*), capuchin (*Sapajus apella, Cebus apella,* and *Cebus libidinosus*), and marmoset (*Callithrix penicillata*) species are the most commonly detected natural hosts among the NHPs [4]. 

### The One Health Framework

The framework of this study was One Health; there are many definitions of One Health, which is considered by some authors as a collaborative effort of multiple disciplines to attain optimal health for humans, animals, and the environment [9,10,11].

However, for this research, the following definition is used: “A transdisciplinary definition of One Health views how animals, humans, and their shared settings or environment (such as ecosystems, soil, climate) are linked and are affected by the socioeconomic interests of humans (such as food production, trade, tourism) and external pressures (such as urbanization, migration, demographics). It also considers how different disciplines can together provide new methods and tools for research and implementation of effective services to support the formulation of norms, regulations, and policies to the benefit of humanity and animals, while considering the environment, for current and future generations. This approach will improve prediction, detection, prevention, and control of infectious hazards and other issues affecting health and well-being in the interface and contribute to the UN’s Sustainable Development Goals to help to improve equity in the world” [12]. 

The aim of this research is to document OROV’s presence in Brazil in humans and select animal species using the One Health approach as well as geospatial techniques. Reviewing the reports with this integrated vision while identifying the different species and biomes in which OROV has been detected throughout the country can help to predict the occurrence of spillover of this disease. Additionally, employing this approach to address OROV in Brazil could raise questions of possible environmental and social drivers of this disease that could be the topics of future studies. This research can also be used to strengthen OROV control efforts in Brazil through integrated surveillance, as well as to facilitate the faster detection of new cases or areas of concern. This is important to reduce the risk of large outbreaks and a potential public health threat of national or international concern.

## 2. Materials and Methods

In order to address the presence of OROV in Brazil, a scoping review of the current literature was conducted following the Preferred Reporting Items for Systematic Reviews and Meta-Analyses or PRISMA statement (Appendix A) [13]. The current literature, constituting articles published from the year 2000 to 2021, was analyzed across the PubMed, SciElo, Web of Science, and Latin American and Caribbean Health Sciences Literature (LILAC) databases. Studies included in this review consist of all reports, such as original research as well as review articles. The uniform search terms “Oropouche virus” and “Brazil” were used across all databases to pull relevant articles. 

Final data gathered from the publications included were the location of the disease presence by state (first subnational level) and municipality (second subnational level, when available), the species of detection, the year that the case occurred, and, if published, the number of cases. It should be noted that the term “species of detection” in this study is used to identify the species in which the case of OROV was detected through laboratory testing. In order to account for the fact that the method of confirmation testing may differ depending on the laboratory performing the study in question, the blanket term of case “detection” was chosen to cover the varying types of serology conducted by the publications included in this review. Due to this, articles which contained reports of both confirmed cases of OROV and those derived from antibody testing, demonstrating a past or resolving infection, have been included. Irrelevant papers were excluded based on whether or not the article was missing one or more of the following: the state in Brazil where OROV was reported, the year that the paper was published, and the availability of the article in full text. This information was then compiled into a collective, summary database for evaluation. The final dataset included OROV reports documented from the years of 1960 to 2018 found in the collected articles published between 2000 and 2021.

After this first round of analysis, a second database was created to geocode the data. Geocoding was meant to place the collected data by location and species. Six species subgroups were established: human, non-human primates (NHPs), midges and mosquitoes, sloths, not identified, and others. The data were then further sectioned out by location of case detection by municipality (second subnational level) and the year of case occurrence. A numerical coding system was used within this database to indicate the presence, or lack thereof, of OROV. The number “1” was used to demonstrate that the Brazilian state and municipality in question reported evidence of this disease within the indicated period and species group, while the code “0” indicated no documentation of presence. 

The geocoded database, also containing corresponding codes to identify location, was then used to map these data in Geographical Information Systems or ArcGIS Pro software from ESRI. Initially, a map illustrating which locations reported OROV’s presence overall was created to outline where the disease was occurring most frequently by first and second subnational level. In an effort to further understand what drives the presence of this disease, data on the biomes of Brazil and their distribution through the country were added to a second map overlapping with OROV case data collected from this review. Biome data were obtained from the World Wildlife Fund’s Terrestrial Ecoregions of the World database [14]. A map was also created to demonstrate the movement of OROV reports throughout Brazil by ten-year increments. The country outline for these maps was supplied from ESRI. Municipality codes as well as the maps’ shapefiles were downloaded from the Brazilian Institute of Geography and Statistics or IBGE [15]. Further, population and GDP per capita data by state were also collected from IBGE [16,17]. 

## 3. Results

A total of 117 articles were reviewed across four databases. From this collection, 41 publications identified natural cases of OROV in Brazil and were therefore considered relevant to the intention of this study. Further, 69 articles were excluded from this review; six of these papers were excluded on the basis of not being available in full text. The step-by-step designation process of reviewed papers followed the Preferred Reporting Items for Systematic Reviews and Meta-Analyses or PRISMA flow diagram (Figure 1). A total of 458 individual cases of OROV contributed to the dataset. 

This review of the current literature uncovered that OROV has been found in half of Brazil’s states (14 of 27 states reported cases). A total of 272 cases of OROV were reported among humans in Brazil between 1960 and 2018. The majority of the human cases were detected in Pará state, located in the Brazilian legal Amazon region. Reports among four genera of non-human primates (NHPs) carrying the disease were found in five separate states and four vectors, including one midge species and three mosquito species, in three states. A list of all species detected is included as well (Appendix A) [1,3,4,5,18,19,20,21,22,23,24,25,26,27,28,29,30,31,32,33,34,35,36,37,38,39,40,41,42,43,44,45,46,47,48,49,50,51,52,53,54]. Interestingly, all 21 cases of OROV in sloths were found only in the state of Pará. Additionally, 97 reports of the virus were among unidentified species, recorded across nine states. Figure 2 outlines the presence of human cases and the four host types selected for the study by state. 

It was found that NHP species are most commonly found in the tropical and subtropical dry broadleaf forests (TSDBF) biome, while midge and mosquito vectors are found in both the TSDBF and the tropical and subtropical moist broadleaf forests (TSMBF) biomes (Figure 2). Cases among sloths were exclusively found in the TSMBF biome (Table 2). 

Data organized by second subnational level found evidence of OROV in 67 municipalities across Brazil. With a total of 5568 municipalities in Brazil, OROV was detected in 1.2% of the country’s municipalities. Of those municipalities, 53 were found in the northern region, indicating that this region has the largest prevalence of cases. Notably, this region is home to the legal Brazilian Amazon area, with a TSMBF biome (Figure 2). The southeast region, which is the most populated region in Brazil, with the highest GDP per capita, has the lowest prevalence of this disease, with only two municipalities reporting evidence of OROV. This region contains the coast of the TSMBF biome as well as part of the tropical grasslands, savannas, and shrublands (TSGSS) biome (Figure 2).

Pará state, located in the northern region, with a population of 8,777,124, was found to report disease presence in 39 of its municipalities (Table 3). Pará holds the largest circulation of cases internally, with approximately 27.1% of its municipalities reporting OROV (Table 3). Meanwhile, Minas Gerais reports the smallest distribution of cases of the states, in 0.23% of its municipalities. Published cases from Amapa and Goiás, in the north and central–west regions, respectively, indicate that OROV has been found in only one of the municipalities in each of these states (Table 3 and Figure 3). Two states, Rio Grande do Sul and Sao Paulo, do not include data by municipality; OROV presence was only collected by state level (Figure 3). 

Reviewing the cases of OROV by the year in which they occurred, the steady expansion of the virus’s geographical boundaries over time can be seen (Figure 4). Prior to 1979, cases of OROV were recorded in the northernmost part of Pará state, among only 22 municipalities. In the 1980s, however, outbreaks occurred in the surrounding states of Amapa, Amazonas, Acre, Maranhao, Rondonia, and Tocantins. During this time, OROV cases also continued to expand within the states that it was previously detected in. For instance, Pará state reported cases in five new, additional municipalities in the 1980s. This trend continued through the 1990s, before OROV was detected in the states of Minas Gerais and Rio Grande do Sul in the 2000s. By the 2010s, cases were being reported in Bahia, Goias, Mato Grosso, Mato Grosso do Sul, and Sao Paulo as well. The first detection of OROV in Brazil in the first time period (before 1979) was found only in Pará state. However, by the last time period (2010s), OROV had been detected in 14 states. 

## 4. Discussion

### 4.1. Drivers Associated with the Occurrence of Outbreaks

To better predict, detect, and respond to possible OROV epidemics, it is important to understand the possible environmental and socioeconomic drivers of this infectious disease [55]. The key factors that drive the spread of OROV can be broken down into two categories: environmental and social. Environmental drivers include: the presence of vectors and reservoir hosts, the state of their natural habitat, the type of biome, ecological pressures, and the presence of OROV in the area. Social drivers are as follows: the type of human activities taking place in the state or municipality within Brazil, the movement of people in and out of the country, and the living conditions of the residents. 

These drivers are interconnected at all times, and changes among them can contribute to the further spread of this disease. For example, land clearing for the purposes of human development, which can ultimately lead to deforestation, alters the natural habitat of the reservoir and vector hosts that carry OROV. This was seen during the 1958 construction of the Belem–Brasilia highway. Extending through the states of Pará, Maranhao, Tocantins, and Goiás, this roadway serves as a catalyst for increased economic development and inter-state travel. However, its creation required a large area of land to be cleared and waterways to be altered [56,57]. Two years later, in 1960, OROV was isolated in both a sloth and a group of mosquitoes located in the areas bordering this construction site [32]. When humans enter the environment with the intention of development, they change the natural ecosystem in place, which can lead to increased close contact among humans and animals, thus facilitating the potential spread of disease [58].

By facilitating a rise in precipitation and warm weather, climate change alters natural weather patterns and fuels more severe storms across the globe. Brazil specifically has experienced devastating flash floods and subsequent landslides in recent years [59]. Flooding can be correlated with the increased incidence of vector-borne diseases including OROV, as it creates standing bodies of water that provide the perfect habitat for mosquitoes to use as breeding sites [60]. By maintaining the population of vector species that spread OROV, such as *Culex quinquefasciatus* or *Ochlerotatus serratus*, for example, the potential for transmission is greatly increased. Vectors such as the *Culicoides paraensis* midge may also be affected by changes in global weather patterns. While they are primarily a tree hole species, flooding can create more habitats for larval development, thus increasing disease transmission [61]. To address this, Brazil’s Ministry of Health currently implements control efforts focused on household larvicide use as well as ultra-low-volume spraying of insecticides for the purposes of epidemic prevention [62].

It is important to consider the movement of people as a potential driver for OROV. For instance, if an individual who has contracted OROV travels to a new region and is bitten by a vector, this midge or mosquito can contract the virus via this blood meal. Once infectious, the vector can transmit OROV to future individuals that it bites during its lifetime, potentially spurring an outbreak in a new location. This will not only maintain the spread of OROV, but also could contribute to expanding the geographic boundaries of its presence within the country.

Poor living conditions and low accessibility to healthcare services can also lead to the persistence of this disease in Brazil. This study found that human cases of OROV were most prevalent among urban settings. As of 2020, 87% of Brazil’s population lives in urban areas [63]. Cities are known to struggle with poor sanitation services, especially in low-income communities. This is common in locations with a high density of people, such as Belem in Pará and Manaus, Amazonas. At 32 and 21 cases, respectively, these municipalities are among the top three highest case counts of OROV in this study, with Belem holding first place. The congestion of people can lead to increased rates of disease transmission as infected vectors in the area pass on the disease to other residents whom they bite during their life cycle.

However, this prevalence of urban-based cases may be attributed to the significant underreporting of OROV cases in rural villages. Universal health coverage (UHC) is provided to every citizen in Brazil through the Sistema Único de Saúde (SUS). Despite this, a study conducted in Brazil’s northeastern region shows that 50% of the rural population in this region resides more than 5 km from their nearest health facility, while 60% of this population lives more than 10 km away [64]. The number of doctors available in public health facilities is, also, comparatively higher among urban locations. Barriers such as these lead to the substantial underreporting of OROV cases among rural communities as they are deterred from seeking medical treatment for their illness. 

Additionally, it should be considered that those who do seek treatment may not be tested for OROV. As several other arboviruses are endemic in Brazil, including dengue virus, physicians will often consider testing for these illnesses first when addressing a patient who is presenting febrile symptoms. Due to this, OROV is often overlooked and not considered as a potential prognosis. For example, in 2016, Bahia state reported five human patients who tested positive for OROV in the municipality of Salvador. Based on their presentation of febrile symptoms, patients initially underwent testing for dengue virus, chikungunya, and Zika virus. However, further diagnostics confirmed that they had contracted OROV. Further research into their past histories revealed that none of the five patients had traveled to locations considered to be endemic for the disease prior to them becoming symptomatic. This cluster is thought to be the first report of OROV in Bahia [21]. 

### 4.2. The Expansion of Cases in Brazil

In addition to the first identified cases in Bahia in 2016, this study has identified reports of OROV in an additional three states located outside of the moist tropical forest biome, including the Brazilian legal Amazon. Given that this is the preferred habitat of the virus’s reservoir and vector host species, it was surprising to find evidence of OROV in the states of Rio Grande do Sul, Goiás, and Sao Paulo as well, where the dry tropical forest biome is prominent.

Between the years of 2002 and 2007, an environmental surveillance program was conducted in Rio Grande do Sul, Brazil. The purpose of this program was to capture and test NHPs found in the area on the basis that they may be carrying arboviruses that represent a risk to human health [47]. Of the NHPs tested, one howler monkey (within the *Alouatta* genus) was positive for OROV antibodies. Additionally, multiple publications report that the municipality of Goiania, Goiás has recently reported cases of OROV [42,44]. 

As human activities, such as land clearing and construction, continue to encroach on Brazil’s natural forests, species are forced to migrate from their natural habitats to new locations. The detection of the mentioned NHPs in Rio Grande do Sul and Goiás indicates that the forced migration of these species could be the case here. This is concerning as vector species could take a blood meal from these NHPs, subsequently giving them the ability to further transmit OROV to future hosts that they may bite, extending the geographical boundaries of this disease. 

In Sao Paulo, during the year 2016, two patients demonstrating febrile symptoms tested positive for OROV via antigen testing [4]. The circumstances of these patients are unknown, and it is unclear whether they traveled to another region prior to being tested for OROV. However, the state of Sao Paulo is well known for high rates of migration, its booming tourism sector, and being home to a well-functioning transportation system that facilitates easy inter-state travel for its residents [4]. 

Another important aspect related to OROV’s emergence as a human viral pathogen of potential public health risk of international concern to be analyzed is its possible ability to be transmitted by urban and peri-urban mosquitoes, commonly found throughout the country. The mosquitoes, such as *Aedes aegypti* and *Aedes albopictus,* that can transmit dengue virus, which is endemic in Brazil, are one of the concerns. One study evaluated the ability of OROV to infect, replicate, and be transmitted by these three anthropophilic and urban species of mosquitoes (*Aedes aegypti*, *Aedes albopictus*, and *Culex quinquefasciatus*) [65]. The authors showed that OROV is able to infect and efficiently replicate when systemically injected in all three species tested, but not when orally ingested. This study provides evidence that OROV is restricted by the midgut barrier of these three major urban mosquito species; however, if this restriction is overcome, the virus could be efficiently transmitted to vertebrate hosts [65]. This poses a great risk for the emergence of permanent urban cycles and the geographic expansion of OROV to other continents.

### 4.3. Limitations to This Study

As the methodology of this study employed the use of a scoping review, all data collected and analyzed through this research are considered to be secondary data. Population size and demographic information was also obtained from secondary sources such as IBGE and Knoema. Given this, the compiled OROV case data may be under- or inaccurately reported. Limited laboratory capacity for case detection should also be considered as a potential impact on the findings of this analysis. Publications included in this review utilized various types of serology to identify cases of OROV according to their resource availability and funding. As the consistency of these tests is not uniform throughout all of the articles collected, the sensitivity and specificity of their methods may differ. Additionally, laboratory technology may have been updated since the completion of earlier studies. Given that this study includes publications dating back to the year 2000, scientific development has likely changed the gold-standard diagnostic test for detecting OROV over this time period. Another limitation was that only publications with full text available were included in this review.

The potential underreporting of OROV cases in Brazil may also limit the findings of this review. The limited capacity of rural hospitals and clinics, as well as the lack of accessibility to residents of low-income neighborhoods, can impact the number of infections detected. Additionally, Brazil is endemic for other vector-borne disease that also present with similar, febrile symptoms. Examples include malaria, Zika virus, and dengue virus. As a result, patients will likely be tested for such diseases prior to OROV becoming a consideration for their prognosis.

## 5. Conclusions

This study demonstrated the presence of OROV in humans and in several animal species; the majority of the municipalities where its presence was documented were in the TSMBF biome (Table 2). By recognizing the interconnection of humans, animals, and the environment and their role in spreading OROV, local-level disease control efforts in Brazil can take on an interdisciplinary approach (Table 4). This encourages integrated surveillance over environmental risk maps with actions across all three components (human, animal, and the environment), and communication among different sectors. 

Prioritizing the accurate and up-to-date reporting of cases throughout the country can allow decision makers to take informed action regarding control measures. Understanding which vectors populate particular locations (by both first and second subnational levels) within Brazil can aid in these informed decisions. These data can also be used to assess the current and future risk that OROV may pose to both Brazil’s urban and rural populations. Further studies of the possible drivers related to OROV’s presence discussed in this descriptive analysis could be performed using GIS and more advanced techniques with open-access socioeconomic and environmental data. 

These efforts are key to not only reduce the incidence of OROV among the population, but also to contain the disease before it spreads to further locations. As the high epidemic potential of this disease has been demonstrated, as well as the fact that much of Brazil’s population lives in close contact with their surrounding environment, the use of the One Health framework to gauge future spillover events of OROV is recommended. 

## Figures and Tables

**Figure 1 tropicalmed-07-00111-f001:**
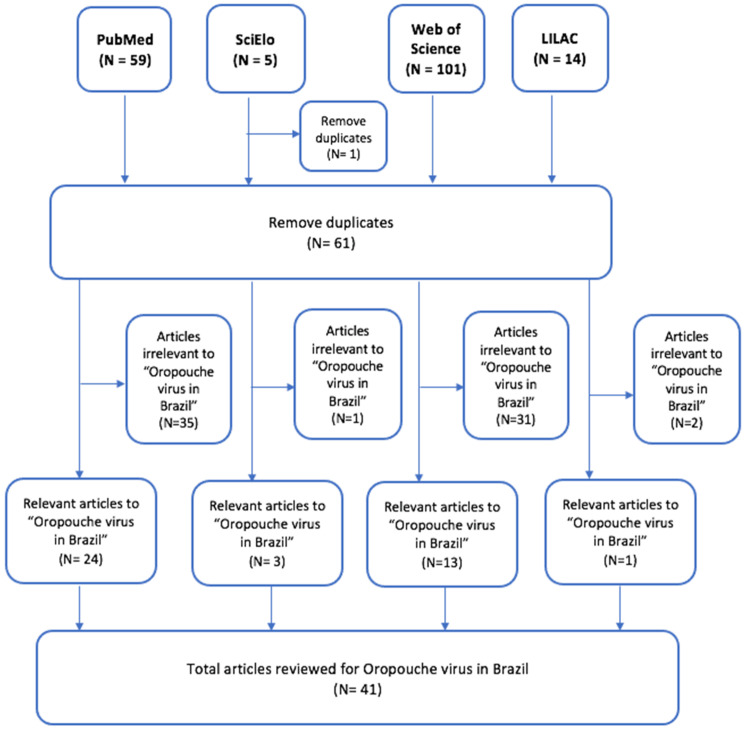
Flow chart of search results for scoping review.

**Figure 2 tropicalmed-07-00111-f002:**
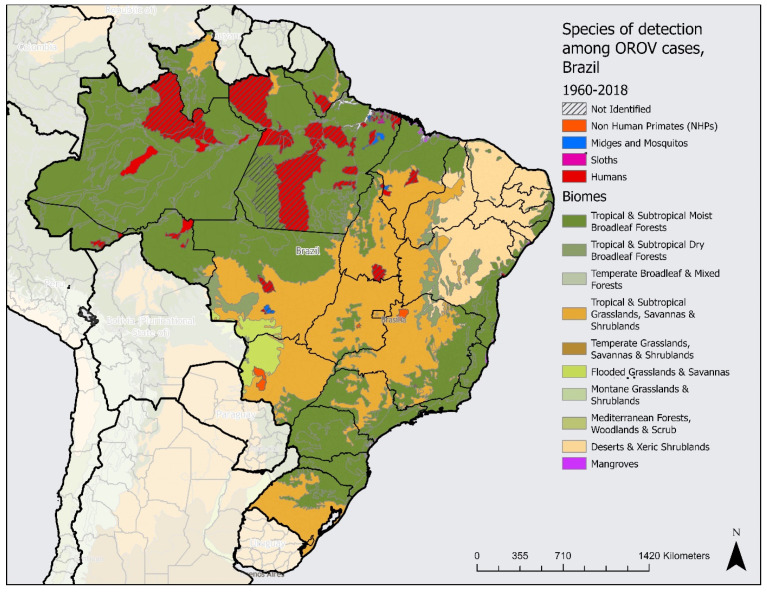
Oropouche virus presence by species over biome, by municipality, Brazil, 1960 to 2018.

**Figure 3 tropicalmed-07-00111-f003:**
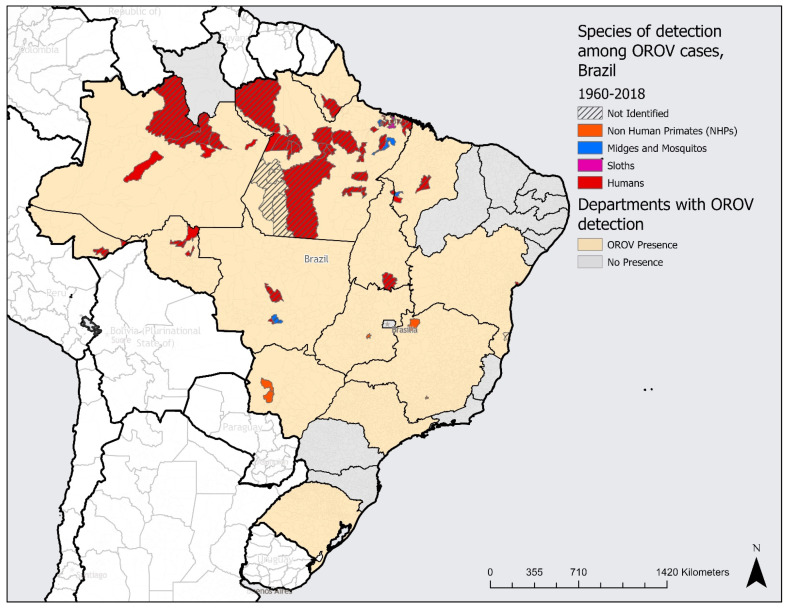
Documented presence of Oropouche virus by species, by state and municipality, Brazil, 1960 to 2018.

**Figure 4 tropicalmed-07-00111-f004:**
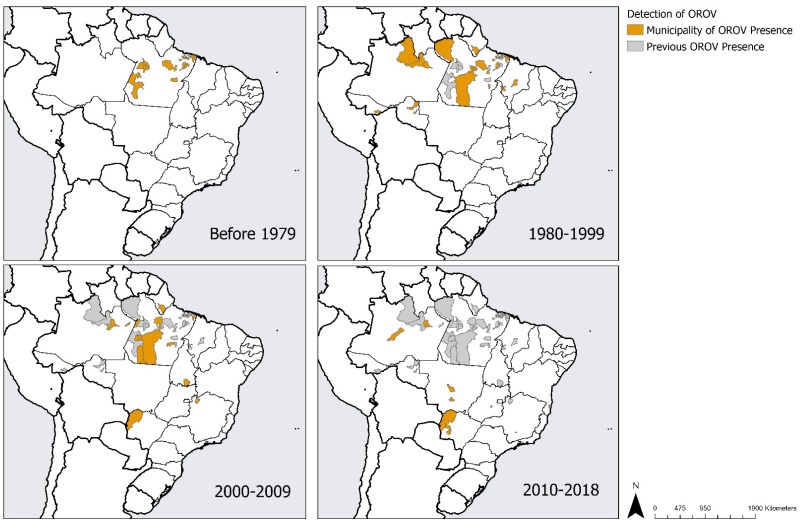
Documented presence of Oropouche virus by 10 years period, Brazil, 1960 to 2018.

**Table 1 tropicalmed-07-00111-t001:** OROV outbreaks with large human case counts in Brazil (1961–2006).

Location *	Year	Case Count
Belem, Pará	1961	11,000
Braganca, Pará	1967	6000
Santarem, Pará	1975	14,000
Belem, Pará	1979–1980	>100,000
Manaus, Amazonas	1980–181	97,000
Ariquemes, Rondonia	1991	94,000
Magalhaes Barata, Pará	2006	17,000

* Outbreak data collected from sources included in this review [1,3,4].

**Table 2 tropicalmed-07-00111-t002:** Evidence of the presence of Oropouche virus by species, biome, and GDP per capita, by first subnational level, Brazil (2000–2021).

First Subnational Level (State)	Humans	NHPs	Midge and Mosquitoes	Sloths	Not Identified	Major Biomes	GDP per Capita
Acre	X				X	TSMBF	1,772,241
Amapa	X				X	TSMBF	2,068,821
Amazonas	X				X	TSMBF	2,610,172
Para	X		X	X	X	TSMBF	2,073,460
Rondonia	X				X	TSMBF	2,649,712
Tocantins	X				X	TSGSS	2,502,180
*North region*	X		X	X	X		
Bahia	X				X	DXS, TSGSS, & TSMBF	1,971,621
Maranhao	X		X		X	TSMBF & TSGSS	1,375,794
*Northeast region*	X		X		X		
Minas Gerais		X				TSGSS & TSMBF	3,079,404
Sao Paulo	*X*					TSGSS & TSMBF	5,114,082
*Southeast region*	X	X					
Rio Grande do Sul		X				TSGSS	4,240,609
*South region*		X					
Goias	X	X				TSGSS	2,973,240
Mato Grosso	X		X		X	TSMBF & TSGSS	4,078,732
Mato Grosso do Sul		X				TSGSS	3,848,283
*Central–west region*	X	X	X		X		
Brazil	X	X	X	X	X		3,516,170

Legend: Tropical and subtropical moist broadleaf forests (TSMBF); Tropical and subtropical grasslands, savannas and shrublands (TSGSS); Deserts and xeric shrublands (DXS).

**Table 3 tropicalmed-07-00111-t003:** Evidence of the presence of Oropouche virus organized by second subnational level within the states of Brazil (2000–2021).

Location (State)	Population	Total Number of Municipalities in the State	Number of Municipalities with Evidence of Presence	Percentage of Municipalities with Evidence of Presence
Acre	906,876	22	2	9.09
Amapa	877,613	16	1	6.25
Amazonas	4,269,995	62	6	9.68
Pará	8,777,124	144	39	27.08
Rondonia	1,815,278	52	3	5.77
Tocantins	1,607,363	139	2	1.43
*North region*	*18,254,249*	*435*	*53*	*12.18*
Bahia	14,985,284	417	2	0.48
Maranhao	7,153,262	217	3	1.38
*Northeast region*	*22,138,546*	*634*	*5*	*0.79*
Minas Gerais	21,411,923	853	2	0.23
*Southeast region*	*21,411,923*	*853*	*2*	*0.23*
Goias	7,206,589	246	1	0.41
Mato Grosso	3,567,234	141	3	2.13
Mato Grosso do Sul	2,839,188	77	3	3.90
*Central–west region*	13,613,011	*464*	7	1.51
Brazil	213,317,639	5,570	67	1.20

**Table 4 tropicalmed-07-00111-t004:** One Health components in Brazil identified from this study.

Human	Animal	Environment
272 human cases of OROV found across 14 states in Brazil Mostly found among the urban population Higher epidemic potential in its urban cycle Migration of people may contribute to its spread to new locations	4 genera of NHPs found among 4 Brazilian states 4 midge and mosquito species identified in 3 states in Brazil 21 cases among sloths were collected in 1 state 5 species classified as “other” were found in 2 states	Predominantly found in the moist, tropical forest biome Amazon Rainforest and Mata Atlântica forest Smallest distribution of cases detected among the desert Exacerbated by deforestation Changing weather patterns

## Data Availability

The Excel database including the summary of all articles in this review is available on request from the corresponding author.

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
