# Peer review of "Presence and Multi-Species Spatial Distribution of Oropouche Virus in Brazil within the One Health Framework"

_tropicalmed, 2022, doi:10.3390/tropicalmed7060111_

Round 1

Reviewer 1 Report

This is simple and well-designed review of publications regarding OROV cases in Brazil. I have no major concerns and only minor suggestions.

Line 20 - The wording "to suggest of" is clumsy and should be revised.

Line 45 - The word "overtime" should be changed to "over time" (2 words).

Line 53 (In Table 1) - The second row for Belem, Para should probably be moved down in the table so the records are in chronological order.

Line 107 - You can remove "was" from the sentence.

Line 133-134 - The S in GIS is typically "Systems" rather than "Services" and the last two words of the sentence should be "from Esri" rather than "on ESRI." Esri is the company that makes the software, not a system that it runs on.

Line 139 - "Wild" should be "Wildlife."

Line 142 - Shapefiles is one word, not two.

Line 171-174 - This feels like two sentences were merged into one. I would recommend dropping "were identified" from the last part of the sentence.

Lines 176-177 - There are more than "four animal species" here. I believe what is meant is four groups or four host types. Same comment for the Figure 2 caption.

Figure 2 - It is practically impossible to tell all of those biome colors apart - I can only distinguish five on the map but there are 10 in the legend. Can any of these be combined or removed to simplify the map?

Line 195 (and throughout) - Para is sometimes printed with an accent (Pará) and other times not. I suggest revising for consistency throughout.

Line 201 - Remove one or the other use of "only" in this sentence.

Line 245 - It may be too presumptuous to say "As a result..." here since this is only correlation. Consider rewording.

Author Response

Letter to reviewers of the manuscript "Multi-species and Spatial Distribution of Oropouche Virus in Brazil Using the One Health Approach" (# 1728374)

We would like to thank the reviewers and editors for their excellent comments in our manuscript and for recognizing its importance. All requests and suggestions were very welcomed, and most have been included in the current version. Please refer below to our responses and the descriptions of the changes made to the manuscript.

Reviewer #1:

This is simple and well-designed review of publications regarding OROV cases in Brazil. I have no major concerns and only minor suggestions.

Thank you for your comments, they are very appreciated!

Line 20 - The wording "to suggest of" is clumsy and should be revised.

We reworded to “Utilizing One Health was important to understand the distribution of OROV across several species and to suggest possible environmental, socioeconomic, and demographic drivers of the virus’s presence” (Page 1, Line 20).

Line 45 - The word "overtime" should be changed to "over time" (2 words).

The sentence was reworded and now reads “Outbreaks of OROV with large case counts have been reported in Brazil since 1961 (Table 1).” (Page 1, Line 45).

Line 53 (In Table 1) - The second row for Belem, Para should probably be moved down in the table so the records are in chronological order.

Table 1 was readjusted to be in chronological order and now appears as seen below (Page 2, Line 54).

Table 1. OROV outbreaks with large human case counts in Brazil (1961 – 2006).

Location*

Year

Case count

Belem, Para

1961

11,000

Braganca, Para

1967

6,000

Santarem, Para

1975

14,000

Belem, Para

1979-1980

>100,000

Manaus, Amazonas

1980-181

97,000

Ariquemes, Rondonia

1991

94,000

Magalhaes Barata, Para

2006

17,000

Line 107 - You can remove "was" from the sentence.

The extra word was removed, and the sentence now reads “Final data gathered from the publications included the location of the disease presence by state (first subnational level) and municipality (second subnational level, when available), the species of detection, the year the case occurred, and, if published, the number of cases” (Page 3, Line 126-129).

Line 133-134 - The S in GIS is typically "Systems" rather than "Services" and the last two words of the sentence should be "from Esri" rather than "on ESRI." Esri is the company that makes the software, not a system that it runs on.

Thank you for your comments, we clarified: “The geocoded database, also containing corresponding codes to identify location, was then used to map this data in Geographical Information Systems or ArcGIS Pro software from ESRI” (Page 3, Line 153-154).

Line 139 - "Wild" should be "Wildlife."

The organization’s name was updated to the “World Wildlife Fund” (Page 3, Line 160).

Line 142 - Shapefiles is one word, not two.

The correction was made, and the sentence now reads “Municipality codes as well as the maps’ shapefiles were downloaded from the Brazilian Institute of Geography and Statistics or IBGE” (Page 4, Line 163-164).

Line 171-174 - This feels like two sentences were merged into one. I would recommend dropping "were identified" from the last part of the sentence.

The sentence was reworded to now read “Reports among four genera of non-human primates (NHPs) carrying the disease were found in five separate states and four vectors, including one midge species and three mosquito species, in three states” (Page 4, Lines 179-181).

Lines 176-177 - There are more than "four animal species" here. I believe what is meant is four groups or four host types. Same comment for the Figure 2 caption.

You are correct, thank you! The sentence has been amended to “Figure 2 outlines the presence of human’s cases and the four host types selected for the study by state” (Page 4, Line 300).

Figure 2 - It is practically impossible to tell all of those biome colors apart - I can only distinguish five on the map but there are 10 in the legend. Can any of these be combined or removed to simplify the map?

The color pallet of our map has been changed. The maps are much easier to follow now!

Line 195 (and throughout) - Para is sometimes printed with an accent (Pará) and other times not. I suggest revising for consistency throughout.

“Para” has been replaced with Pará throughout our paper now.

Line 201 - Remove one or the other use of "only" in this sentence.

The extra word has been deleted and the sentence now reads “Published cases from Amapa and Goiás, from the North and Central-West region respectively, indicate that OROV has been found in only one of the municipalities in each of these states (Table 2 and Figure 3)” (Page 6, Line 345).

Line 245 - It may be too presumptuous to say "As a result..." here since this is only correlation. Consider rewording.

This sentence has been amended to now say “Two years later in 1960, OROV was isolated in both a sloth and a group of mosquitoes located in the areas bordering this construction site” (Page 10, Line 446).

Reviewer 2 Report

With this paper, authors characterize the distribution of OROV within human and non-human animal hosts and vectors between 1960 and 2018 in Brazilian states. They make mention of using a One Health approach and geospatial techniques to describe the distribution of OROV, but these approaches involved no real analysis of patterns, and are purely used to speculate on the potential drivers of observed patterns. The Discussion section makes up more than half of the paper, and relies primarily on conjecture regarding how social and environmental factors might be associated with OROV distribution. There is no data analysis to back any of these statements. Moreover, although authors state that they made use of a One Health approach, it's unclear what social factors they actually accounted for. This "One Health approach" was really only utilized in the Discussion section when authors speculated on what factors (within a One Health framework) might contribute to observed patterns of OROV. 

The following are strengths of the work provided by authors: 

(1) Comprehensive literature search and review of studies, which allowed for compilation of OROV data in multiple hosts

(2) Use of the PRISMA framework for reporting literature review methods and results

(3) A great discussion of One Health concepts as they relate to OROV in Brazil

The following are some of the limitations of the work provided by authors: 

(1) Lack of a testable hypothesis regarding social and environmental drivers of OROV

(2) No compilation or use of ancillary data (except for the biome data) relevant to potential social and environmental drivers of OROV (e.g. state-level socioeconomic measures and population data, satellite imagery data related to deforestation/forest cover, flooding, precipitation, and temperature extremes)

(3) No statistical analysis to identify potential drivers of OROV (e.g. generalized estimating equations with a logit link and state-level clusters)

Overall, although the research objective addressed in this paper is a worthy and interesting one, the lack of an analytical framework with which to identify environmental and social drivers of OROV occurrence is too much of a limitation to recommend this paper for publication. 

Below are some additional suggestions for improvements to the paper.

Title: Consider omitting "Using the One Health Approach" from the title, since this was really only addressed in the Introduction and Discussion sections.

Page 2, lines 86-87: Paraphrase rather than use direct quotes here. 

Section 1.1: Because no analysis was conducted regarding environmental and social drivers of OROV patterns, this mention of the One Health approach should be limited to the Discussion setion. Alternatively, if authors want to keep One Health at the forefront, then there should be formal analysis of environmental and social drivers of OROV occurrence. 

Page 3, line 89 -- "using the One Health approach": This term is not explicitly defined here, and remains somewhat vague. Consider stating what the One Health approach actually entails with respect to this paper. 

Page 3, lines 118-119 -- exclusion criteria for literature review: The criterion associated with availability of the full-text article results in a major limitation of the study, and it is unclear to what extent this occurred based on the flowchart provided in Figure 1. 

Table 2: Authors should consider adding another column that cites the studies from which these data were summarized, along with the human and non-human animal hosts and vectors in which OROV was detected. 

Discussion section: Authors should consider including an additional limitation in the Discussion section related to the case data. Cases reported in a given state may not represent where infections were acquired. 

Author Response

Letter to reviewers of the manuscript "Multi-species and Spatial Distribution of Oropouche Virus in Brazil Using the One Health Approach" (# 1728374)

We would like to thank the reviewers and editors for their excellent comments in our manuscript and for recognizing its importance. All requests and suggestions were very welcomed, and most have been included in the current version. Please refer below to our responses and the descriptions of the changes made to the manuscript.

Reviewer #2:

With this paper, authors characterize the distribution of OROV within human and non-human animal hosts and vectors between 1960 and 2018 in Brazilian states. They make mention of using a One Health approach and geospatial techniques to describe the distribution of OROV, but these approaches involved no real analysis of patterns, and are purely used to speculate on the potential drivers of observed patterns. The Discussion section makes up more than half of the paper and relies primarily on conjecture regarding how social and environmental factors might be associated with OROV distribution. There is no data analysis to back any of these statements. Moreover, although authors state that they made use of a One Health approach, it's unclear what social factors they actually accounted for. This "One Health approach" was really only utilized in the Discussion section when authors speculated on what factors (within a One Health framework) might contribute to observed patterns of OROV. 

The following are strengths of the work provided by authors: 

(1) Comprehensive literature search and review of studies, which allowed for compilation of OROV data in multiple hosts

(2) Use of the PRISMA framework for reporting literature review methods and results

(3) A great discussion of One Health concepts as they relate to OROV in Brazil

The following are some of the limitations of the work provided by authors: 

(1) Lack of a testable hypothesis regarding social and environmental drivers of OROV

(2) No compilation or use of ancillary data (except for the biome data) relevant to potential social and environmental drivers of OROV (e.g. state-level socioeconomic measures and population data, satellite imagery data related to deforestation/forest cover, flooding, precipitation, and temperature extremes)

(3) No statistical analysis to identify potential drivers of OROV (e.g. generalized estimating equations with a logit link and state-level clusters)

Overall, although the research objective addressed in this paper is a worthy and interesting one, the lack of an analytical framework with which to identify environmental and social drivers of OROV occurrence is too much of a limitation to recommend this paper for publication. 

Below are some additional suggestions for improvements to the paper.

Thank you for your comments, they are very appreciated!

Title: Consider omitting "Using the One Health Approach" from the title, since this was really only addressed in the Introduction and Discussion sections.

In order to clarify the One Health approach in our paper we have included an additional table indicating in which species and Brazilian state the disease was identified as well as information on the environment (major biome) and the social factors (GDP per capita) of each state (Table 3). Additionally, population data has been included in Table 2.

As per another reviewer’s recommendation, we have also changed the title of our paper to “Multi-species and Spatial Distribution of Oropouche Virus in Brazil Within the One Health Framework.”

Page 2, lines 86-87: Paraphrase rather than use direct quotes here. 

We have changed the order of the citations and included the quote of the One Health definition of one of the coauthors in the Oxford Encyclopedia of Global Public Health [12]. Lines 86 to 87 has been paraphrased and the section has been written.

Section 1.1 now reads “The framework of this study was One Health, there are many definitions of One Health, which is considered by several authors as a collaborative effort of multiple disciplines to attain optimal health for humans, animals, and the environment [9, 10, 11].

However, for this research, we use the following definition “A transdisciplinary definition of One Health views how animals, humans, and their shared settings or environment (such as ecosystem, soil, climate) are linked and are affected by the socioeconomic interest of humans (such as food production, trade, tourism) and external pressures (such as urbanization, migration, demographics). It also considers how different disciplines can together provide new methods and tools for research and implementation of effective services to support the formulation of norms, regulations, and policies to the benefit of humanity and animals, while considering the environment, for current and future generations. This approach will improve prediction, detection, prevention, and control of infectious hazards and other issues affecting health and well-being in the interface and contribute to the UN’s Sustainable Development Goals to help to improve equity in the world” [12].

Section 1.1: Because no analysis was conducted regarding environmental and social drivers of OROV patterns, this mention of the One Health approach should be limited to the Discussion section. Alternatively, if authors want to keep One Health at the forefront, then there should be formal analysis of environmental and social drivers of OROV occurrence. 

We agree with the reviewer that it would be excellent if we could go deeper into the statistics and use more sophisticated GIS techniques to analyze the possible drivers; however, the aim of this research was “to document OROV’s presence in Brazil in humans and select animal species using the One Health approach as well as geospatial technics.” We document cases in humans, as well in nonhuman primates, sloths, midges and mosquitoes and we map its presence over biomes. We did a descriptive analysis of the presence of OROV over biome. But for this study, we do not have the ambition to do statistical analysis using regression involving downloading geocoded data from different sources. Our aim was only to describe the possible patterns, as we did. However, we recognize that through the way we have written the research aim it could be understood that we will do much more advanced analysis. We have reviewed the aim of the research as below:

“The aim of this research is to document OROV’s presence in Brazil in humans and select animal species using the One Health approach as well as geospatial technics. Reviewing the reports with this integrated vision, identifying the different species that OROV was detected and in each biome, this occurred throughout the country can help predict the occurrence of future spillover events of this disease as it spreads from its animal to human hosts. Additionally, employing this approach to address Oropouche virus in Brazil could raise questions of possible environmental and social drivers of this disease that could be topics of future studies” (Page 3, Line 168-174).

To address this comment, we included an extra table with a summary of the appendix table by state describing the presence of different species (human and animals) and included the major biome of the state and GDP per capita (Table 3). This was added to the manuscript to make the multi-species aspect of the study more visible.

Page 3, line 89 -- "using the One Health approach": This term is not explicitly defined here and remains somewhat vague. Consider stating what the One Health approach actually entails with respect to this paper. 

As per your initial recommendation, we have restructured section 1.1 of our manuscript which addresses the definition of One Health and our use of the framework within this paper. Here, we have included an additional definition of this approach to further touch on its purpose and what one health entails. Please find the full definition below for your reference.

“A transdisciplinary definition of One Health views how animals, humans, and their shared settings or environment (such as ecosystem, soil, climate) are linked and are affected by the socioeconomic interest of humans (such as food production, trade, tourism) and external pressures (such as urbanization, migration, demographics). It also considers how different disciplines can together provide new methods and tools for research and implementation of effective services to support the formulation of norms, regulations, and policies to the benefit of humanity and animals, while considering the environment, for current and future generations. This approach will improve prediction, detection, prevention, and control of infectious hazards and other issues affecting health and well-being in the interface and contribute to the UN’s Sustainable Development Goals to help to improve equity in the world” [12].

Page 3, lines 118-119 -- exclusion criteria for literature review: The criterion associated with availability of the full-text article results in a major limitation of the study, and it is unclear to what extent this occurred based on the flowchart provided in Figure 1. 

This concern has now been further addressed through an additional sentence included in the results. “Further, 69 articles were excluded from this review; six of these papers were excluded on the basis of not being available in full text” (Page 4, Line 281).

Table 2: Authors should consider adding another column that cites the studies from which these data were summarized, along with the human and non-human animal hosts and vectors in which OROV was detected. 

Thank you for your comment. This has now been addressed in Appendix A. References have been included by the state in which the case of OROV was detected. Please find this information now in Table A1.

Discussion section: Authors should consider including an additional limitation in the Discussion section related to the case data. Cases reported in a given state may not represent where infections were acquired. 

Section 4.3 has been adjusted accordingly and now addresses this limitation as well (Page 12, Line 630).

Reviewer 3 Report

This review is about the presence and spatial distribution of the Oropouche virus in humans and other animal species in Brazil. 

Abstract: Please write the results in a concise way in the abstract. In the present abstract, the findings are not written clearly.

Introduction: It is well written.

Methodology: Figure 1 text is not clearly visible, so please change it with a good quality figure. Furthermore, in figure 1 at the end, the authors mentioned the review articles for the Oropouche virus in Brazil (n=41). It is not clear whether they included only review articles for this systematic review or all reports, research articles, and others. If these are total articles they reviewed and included in the study then they need to mention "Total articles" or "studies included in the review" as mentioned in the PRISMA flow diagram" in supplementary data.

Results: Adding graphs to show the Oropouche virus prevalence in various species from1960 to 2018 will help the readers to understand the text/content quickly. 

Discussion: It is too long, should be succinct. 

Conclusion: Authors did not conclude the study at the end. They should write the conclusion separately to summarize the findings with future prospects of the study. 

Author Response

Letter to reviewers of the manuscript "Multi-species and Spatial Distribution of Oropouche Virus in Brazil Using the One Health Approach" (# 1728374)

We would like to thank the reviewers and editors for their excellent comments in our manuscript and for recognizing its importance. All requests and suggestions were very welcomed, and most have been included in the current version. Please refer below to our responses and the descriptions of the changes made to the manuscript.

Reviewer #3:

This review is about the presence and spatial distribution of the Oropouche virus in humans and other animal species in Brazil. 

Abstract: Please write the results in a concise way in the abstract. In the present abstract, the findings are not written clearly.

The abstract has been amended accordingly, thank you (Page 1, Lines 17-19).

Introduction: It is well written.

Thank you for your comments!

Methodology: Figure 1 text is not clearly visible, so please change it with a good quality figure.

The resolution of Figure 1 has been updated and the improved version included. The figure is much easier to read now!

Furthermore, in figure 1 at the end, the authors mentioned the review articles for the Oropouche virus in Brazil (n=41). It is not clear whether they included only review articles for this systematic review or all reports, research articles, and others. If these are total articles they reviewed and included in the study then they need to mention "Total articles" or "studies included in the review" as mentioned in the PRISMA flow diagram" in supplementary data.

The final 41 articles include all reports, research articles, etc. As per your comment, we have clarified in the methodology by including the following sentence “Studies included in this review include all reports such as original research as well as review articles” (Page 3, Line 184). Further, figure 1 was amended to say “Total articles reviewed for Oropouche Virus in Brazil (N=41).

Results: Adding graphs to show the Oropouche virus prevalence in various species from 1960 to 2018 will help the readers to understand the text/content quickly. 

In our results section we have included a new table (Table 3) that includes an overall summary of our data by state. Table 3 includes collected data on the species of detection for OROV in Brazil from 1960 to 2018, the major biome, and the GDP per capita of the corresponding state.

We have also included an additional graph in Appendix A demonstrating which species of detection were identified in which states of Brazil (Figure A2).

Discussion: It is too long, should be succinct. 

Please note that the discussion section has been shortened and is now only 2 and a half pages in length.

Conclusion: Authors did not conclude the study at the end. They should write the conclusion separately to summarize the findings with future prospects of the study. 

A conclusion section has been added to summarize our overall findings (Page 12, Line 668-688).

Reviewer 4 Report

In this manuscript, the authors provide a review of the distribution of Oropouche virus in humans and vectors/reservoir hosts in Brazil over the last 20 years. Oropouche is an emerging viral febrile illness primarily transmitted by Culicoides biting midges. Overall, Oropouche virus and its primary vector are relatively understudied, though there is clearly potential for outbreaks to occur. I particularly appreciated that the authors have taken socio-environmental approach to discussing Oropouche emergence in Brazil, as many of these factors are critical to consider, but often overlooked. I have a few comments below for the authors to consider to improve the manuscript.

Editorial Comments:

Line 29: Remove “in their country” from this sentence

Line 36 (and throughout): Para should be Pará, please correct throughout the manuscript

Line 44 (and throughout): In general, and where possible, tables and figures should be referenced in the text after the appropriate section (e.g. (table 1)) rather than specifically described, as is done here

Line 46: Suggest changing “…outbreaks leaving a trail of…” to “…outbreaks resulting in…”

Line 49 (and throughout): “Dengue” should not be capitalized, change “dengue fever” to “dengue virus” when referencing the pathogen and not the disease

Line 61 (and throughout): “Oropouche virus” should be consistently abbreviated to OROV after the first usage

Line 67: Change “wild” to “sylvatic”

Line 70-72: This sentence reads as if only mosquitoes transmit OROV (rather than that only females are involved in pathogen transmission generally). Suggest rewording it to avoid confusion about which species can transmit this pathogen.

Line 72-74: Although you state that human-to-human transmission does not occur, this sentence makes it sound like that’s exactly what happens (“transmitted between infected individuals and healthy individuals”). Suggest rewording.

Line 76: Change “…non-human primates or NHPs…” to “…non-human primates (NHPs)…”

Line 77: Your review didn’t find that these species were reservoir hosts, the studies you cite did. Suggest rephrasing this sentence.

Line 107-108: Change “…included was the location of the disease presence by state…” to “included were the location of the disease by state…”

Line 133: Include company/manufacturer information as well as software version as appropriate

Line 139: Should this be World Wildlife Fund?

150: Report or article? To me, report sounds like an individual disease case. Is that what you mean here?

Line 177: Change “human’s” to “human”

Line 188-190: The second half of this sentence is redundant, suggest rephrasing

Line 195: Awkward sentence, consider rephrasing

Line 355: Change “dengue fever” to “dengue virus”

General Comments:

Line 70: Would be useful to provide a better description of Culicoides, including how their ecology differs from mosquitoes. In general, I felt that the vectors, particularly C. paraensis were glossed over in this review. I would suggest that the authors spend some time describing the roles of C. paraensis vs. mosquitoes in OROV transmission. The authors essentially only talk about mosquitoes, however, C. paraensis is likely the primary vector in the urban, epidemic cycle. The biology of this species is pretty distinct compared to Cx. quinquefasciatus or Ae. aegypti, and will affect the spatial epidemiology of the disease, as well as efforts to control it. In my opinion, this is the weakest point of the manuscript and deserves far more attention.

Line 89: It's unclear what “using the One Health” approach means in the context of assembling this review. I would suggest that it may be more accurate to state that you want to discuss the results of the review in a One Health context, or something similar.

Line 102: Can you briefly describe the PRISMA method for compiling reviews? Or is there a reference that can be cited here?

Lines 250-257: Here you state that flooding events could create vector habitat, however, C. paraensis is primarily a tree hole species. Could the authors perhaps elaborate on how climate change might affect this particular species? I doubt flooding would actually have much of an effect in this case, except maybe in the sylvatic cycle.

Lines 258-270: I’m not sure about this, but is there any work on the role of human movement on dengue or Japanese encephalitis outbreaks (or similar) that could be cited here?

Lines 271-278: This paragraph runs the risk of blaming refugees for OROV outbreaks. I would suggest reframing it as that refugees are a high risk population often forced to live in areas that are at high risk of transmission.

How many of the human case reports included vector surveillance? On your maps, there’s only a few municipalities where OROV is detected in vectors. Is this an area you can identify as a need for controlling/understanding OROV transmission? Is it possible to differentiate areas where surveillance was done and the pathogen wasn’t detected in vectors vs. where it wasn’t looked for?

Figures: Colors on the maps are hard to distinguish when printed in black and white, especially the different biomes

Table 2: Would suggest adding the percent of municipalities with OROV in parentheses in the 3rd column

Author Response

Letter to reviewers of the manuscript "Multi-species and Spatial Distribution of Oropouche Virus in Brazil Using the One Health Approach" (# 1728374)

We would like to thank the reviewers and editors for their excellent comments in our manuscript and for recognizing its importance. All requests and suggestions were very welcomed, and most have been included in the current version. Please refer below to our responses and the descriptions of the changes made to the manuscript.

Reviewer #4:

In this manuscript, the authors provide a review of the distribution of Oropouche virus in humans and vectors/reservoir hosts in Brazil over the last 20 years. Oropouche is an emerging viral febrile illness primarily transmitted by Culicoides biting midges. Overall, Oropouche virus and its primary vector are relatively understudied, though there is clearly potential for outbreaks to occur. I particularly appreciated that the authors have taken socio-environmental approach to discussing Oropouche emergence in Brazil, as many of these factors are critical to consider, but often overlooked. I have a few comments below for the authors to consider to improve the manuscript.

Editorial Comments:

Line 29: Remove “in their country” from this sentence

The phrase has been removed and the sentence now reads “Since this initial case in 1955, Argentina, Bolivia, Brazil, Columbia, Ecuador, Panama, Peru, and Venezuela have all reported the presence of OROV” (Page 1, Line 29).

Line 36 (and throughout): Para should be Pará, please correct throughout the manuscript

“Para” now reads “Pará” throughout our paper.

Line 44 (and throughout): In general, and where possible, tables and figures should be referenced in the text after the appropriate section (e.g. (table 1)) rather than specifically described, as is done here

This change has been made in many locations throughout the paper, thank you for your comment. For example, this can be seen in our update of Lines 60-61, “OROV then rapidly spread to the neighboring states of Amazonas and Rondonia before reaching Mato Grosso state in the 2010s.”

Line 46: Suggest changing “…outbreaks leaving a trail of…” to “…outbreaks resulting in…”

The sentence has been reworded to say “Between the years of 1961 to 1975 the state of Pará experienced four major outbreaks resulting in over 131,000 cases in just 14 years” (Page 2, Line 59).

Line 49 (and throughout): “Dengue” should not be capitalized, change “dengue fever” to “dengue virus” when referencing the pathogen and not the disease

Thank you! This change has been made in Line 62 and throughout our paper.

Line 61 (and throughout): “Oropouche virus” should be consistently abbreviated to OROV after the first usage

The abbreviation “OROV” has been used to indicate Oropouche virus throughout the paper.

Line 67: Change “wild” to “sylvatic”

The sentence had been amended to “OROV is maintained in nature through two distinct cycles: an urban and sylvatic cycle” (Page 2, Line 81).

Line 70-72: This sentence reads as if only mosquitoes transmit OROV (rather than that only females are involved in pathogen transmission generally). Suggest rewording it to avoid confusion about which species can transmit this pathogen.

We have clarified this section to include a more in-depth discussion of OROV’s vectors (both midge and mosquito species). We also specifically changed this sentence. It now reads as follows: “Notably, only adult, female midges are responsible for pathogen transmission. This is because only female vectors take blood meals to support egg production and maturation [4]” (Page 2, Line 85-87).

Line 72-74: Although you state that human-to-human transmission does not occur, this sentence makes it sound like that’s exactly what happens (“transmitted between infected individuals and healthy individuals”). Suggest rewording.

Thank you for your suggestion. This section now reads “Additionally, Culicoides midges are considered to be of public health concern globally as they are a known vector species of several other arboviruses (such as Equine encephalitis and Schmallenberg virus) and approximately 96% of these midge species take blood meals from humans and wild mammals [4]. This is of importance as, in the urban cycle, OROV can be transmitted to individuals who are susceptible to the bite of an infected vector during the blood meal [3]” (Page 2, Line 90-93).

Line 76: Change “…non-human primates or NHPs…” to “…non-human primates (NHPs)…”

This change has been made. The sentence now reads “In its sylvatic cycle, mammals such as non-human primates (NHPs) and sloths (Bradypus tridactylus) act as potential reservoir hosts for OROV” (Page 2, Line 99).

Line 77: Your review didn’t find that these species were reservoir hosts, the studies you cite did. Suggest rephrasing this sentence.

In order to clarify this the sentence was reworded and a citation has been added to the end of Line 100 on Page 2. “It has been found that howler monkey (Alouatta caraya and Alouatta guariba clamitans), capuchin (Sapajus apella, Cebus apella, and Cebus libidinosus), and marmoset (Callithrix penicillata) species are the most commonly detected natural hosts among the NHPs [4].”

Line 107-108: Change “…included was the location of the disease presence by state…” to “included were the location of the disease by state…”

The sentence has been amended to “Final data gathered from the publications included were the location of the disease presence by state (first subnational level) and municipality (second subnational level, when available), the species of detection, the year the case occurred, and, if published, the number of cases” (Line 230).

Line 133: Include company/manufacturer information as well as software version as appropriate

Thank you for your comments, we clarified: “The geocoded database, also containing corresponding codes to identify location, was then used to map this data in Geographical Information Systems or ArcGIS Pro software from ESRI” (Line 310).

Line 139: Should this be World Wildlife Fund?

Yes, thank you! This now says “The World Wildlife Fund.”

150: Report or article? To me, report sounds like an individual disease case. Is that what you mean here?

Yes, that is what was intended to say here. To clarify we have reworded the sentence to “A total of 458 individual cases of OROV contributed to the dataset” (Line 329).

Line 177: Change “human’s” to “human”

This change has been made, thank you.

Line 188-190: The second half of this sentence is redundant, suggest rephrasing

The sentence was rephrased to “With a total of 5568 municipalities in Brazil, OROV was detected in 1.2% of the country’s municipalities. Of those municipalities, 53 were found in the North region indicating that this region has the largest prevalence of cases” (Line 388)

Line 195: Awkward sentence, consider rephrasing

The sentence was changed to “This region contains the coast of the TSMBF biome as well as part of the Tropical Grasslands, Savannas, and Shrublands (TSGSS) biome (Figure 2)." Line 393

Line 355: Change “dengue fever” to “dengue virus”

This change was noted and included throughout the paper.

General Comments:

Line 70: Would be useful to provide a better description of Culicoides, including how their ecology differs from mosquitoes. In general, I felt that the vectors, particularly C. paraensis were glossed over in this review. I would suggest that the authors spend some time describing the roles of C. paraensis vs. mosquitoes in OROV transmission. The authors essentially only talk about mosquitoes, however, C. paraensis is likely the primary vector in the urban, epidemic cycle. The biology of this species is pretty distinct compared to Cx. quinquefasciatus or Ae. aegypti, and will affect the spatial epidemiology of the disease, as well as efforts to control it. In my opinion, this is the weakest point of the manuscript and deserves far more attention.

Further discussion on the vectors of OROV has been added to Line 134 -140 as well as in the discussion section in Line 553.

Additionally, in our research we have set six subgroups of identified host species of OROV. One of these groups includes the “Midge and Mosquito” group. As these two host species are set in the same category, we have amended sentences throughout the manuscript from simply saying “vectors” or “mosquitos” to include both “midge and mosquitoes” for clarity.

Line 89: It's unclear what “using the One Health” approach means in the context of assembling this review. I would suggest that it may be more accurate to state that you want to discuss the results of the review in a One Health context, or something similar.

We have amended the title of our manuscript to now be “Multi-species and Spatial Distribution of Oropouche Virus in Brazil Within the One Health Framework.”

Line 102: Can you briefly describe the PRISMA method for compiling reviews? Or is there a reference that can be cited here?

Thank you, we have included a citation here.

Lines 250-257: Here you state that flooding events could create vector habitat, however, C. paraensis is primarily a tree hole species. Could the authors perhaps elaborate on how climate change might affect this particular species? I doubt flooding would actually have much of an effect in this case, except maybe in the sylvatic cycle.

A deeper consideration of the C. paraensis and how their transmission of OROV is impacted by climate change has been added to Line 553-558.

Lines 258-270: I’m not sure about this, but is there any work on the role of human movement on dengue or Japanese encephalitis outbreaks (or similar) that could be cited here?

Thank you for your suggestion. We have significantly reduced our discussion of human movement and migrant populations among our discussion section. However, we have added a note of the midge vector’s capability of spreading other arboviruses such as Equine Encephalitis virus (EEV) and Schmallenberg virus in Lines 135-137 of our introduction.

Lines 271-278: This paragraph runs the risk of blaming refugees for OROV outbreaks. I would suggest reframing it as that refugees are a high-risk population often forced to live in areas that are at high risk of transmission.

Thank you. This section has now been removed from our discussion section.

How many of the human case reports included vector surveillance? On your maps, there’s only a few municipalities where OROV is detected in vectors. Is this an area you can identify as a need for controlling/understanding OROV transmission? Is it possible to differentiate areas where surveillance was done and the pathogen wasn’t detected in vectors vs. where it wasn’t looked for?

Additional mentions of Brazil’s current vector control efforts have been included in the discussion section (Line 553). We have also included recommendations on using this data to aid in local level decision making regarding vector control moving forward (Line 773).

“While they are primarily a tree hole species, flooding can create more habitats for larval development, thus increasing disease transmission [23]. To address this, Brazil’s Ministry of Health currently implements control efforts focused on household larvicide use as well as ultra-low volume spraying of insecticides for the purposes of epidemic prevention [24].” (Line 553-558).

“Prioritizing accurate and up to date reporting of cases throughout the country can allow local level decision makers to take informed action regarding control measures. Understanding which vectors populate particular locations (by both first and second subnational level) within Brazil can aid in these informed decisions.” (Line 773-778).

Figures: Colors on the maps are hard to distinguish when printed in black and white, especially the different biomes

The color pallet of our maps has been updated. The maps are much easier to follow now!

Table 2: Would suggest adding the percent of municipalities with OROV in parentheses in the 3rd column

The percentages of have been added to an additional column in Table 2, thank you.